# Position: Stop Anthropomorphizing Intermediate Tokens as Reasoning/Thinking Traces!

**Subbarao Kambhampati** [1]  **Karthik Valmeekam** [1]  **Siddhant Bhambri** [1]  **Vardhan Palod** [1]  **Lucas Saldyt** [1]
**Kaya Stechly** [1]  **Soumya Rani Samineni** [1]  **Durgesh Kalwar** [1]  **Upasana Biswas** [1]

## Abstract

Intermediate token generation (ITG), where a model produces output before the solution, has become a standard method to improve the performance of language models on reasoning tasks. These intermediate tokens have been called "reasoning traces" or even "thinking traces" – implicitly anthropomorphizing the traces, and implying that these traces resemble steps a human might take when solving a challenging problem, and as such can provide an interpretable window into the operation of the model's thinking process to the end user. In this position paper, we present evidence that this anthropomorphization isn't a harmless metaphor, and instead is quite dangerous – it confuses the nature of these models and how to use them effectively, and leads to questionable research. We call on the community to avoid such anthropomorphization of intermediate tokens.

## 1. Introduction

Recent advances in general planning and problem solving have been spearheaded by so-called "Long Chain-of-Thought" models, most notably DeepSeek's R1 (Guo et al., 2025). These transformer-based large language models are further post-trained on verifier assisted synthetic problem instances, using iterative fine-tuning and reinforcement learning methods (Kambhampati et al., 2025). Following the now-standard teacher-forced pre-training, instruction fine-tuning, and preference alignment stages, they undergo additional training on reasoning tasks: at each step, the model is presented with a question; it generates a sequence of intermediate tokens (colloquially or perhaps fancifully called a "Chain of Thought" or "reasoning trace"); and it ends it with a specially delimited answer sequence. After verification

of this answer sequence by a formal system, the model's parameters are updated so that it is more likely to output sequences that end in correct answers and less likely to output those that end in incorrect answers with no guarantees of trace correctness.

While (typically) no direct optimization pressure is applied to the intermediate tokens (Baker et al., 2025; Zhou et al., 2025), empirically it has been observed that language models perform better on many domains if they are trained to output such tokens first (Nye et al., 2021; Wei et al., 2022; Zhang et al., 2022; Hsieh et al., 2023; Gu et al., 2023; Guo et al., 2025; Pfau et al., 2024; Muennighoff et al., 2025; Li et al., 2025a). While the fact of the performance increase is well-known, the reasons for it are less clear. Much of the previous work has framed intermediate tokens in wishful anthropomorphic terms, claiming that these models are "thinking" before outputting their answers (Gandhi et al., 2025; Guo et al., 2025; Yang et al., 2025; Zhou et al., 2025; Bubeck et al., 2023). The traces are thus seen both as giving insights to the end users about the solution quality, and capturing the model's "thinking effort."

**In this paper, we take the position that anthropomorphizing intermediate tokens as reasoning/thinking traces is (1) wishful (2) has little concrete supporting evidence (3) engenders false confidence and (4) may be pushing the community into fruitless research directions.** We support our position by collating significant body of emerging work, including that from our group, questioning the interpretation of intermediate tokens as reasoning/thinking traces (Section 4). In Section 5, we will consider alternative views–that include expecting or hoping that intermediate tokens would give end users visibility into the operation of the model, and discuss how they affect our position. Finally, in Section 6, we will provide a call to action for the community that arises naturally from our position.

Anthropomorphization has long been a contentious issue in AI research (McDermott, 1976), and LLMs have certainly increased our anthropomorphization tendencies (Ibrahim & Cheng, 2025). While some forms of anthropomorphization can be treated rather indulgently as harmless and metaphorical, our view is that viewing ITG as reasoning/thinking is

[1] School of Computing & AI, Arizona State University. Correspondence to: Subbarao Kambhampati <rao@asu.edu>.

*Proceedings of the $43^{rd}$ International Conference on Machine Learning*, Seoul, South Korea. PMLR 306, 2026. Copyright 2026 by the author(s).

more serious and may give a false sense of model capability and correctness.

The rest of the paper is organized as follows: We will start in Section 2 by giving some background on the main ideas behind reasoning models, with special attention to post-training on derivational traces.[1] In Section 3, we will discuss the evidence for and ramifications of anthropomorphizing intermediate tokens as reasoning traces. In Section 4, we directly consider the question of whether intermediate tokens can be said to have any formal or human-interpretable semantics. We shall also look at the pitfalls of viewing intermediate tokens as computation that is adaptive to problem complexity. In Section 5, we will consider alternative views, and discuss how they affect our position. Finally, in Section 6, we discuss the implications of our position and issue a call to action for the community.

Before going forward, we should clarify some potential confusion regarding the "reasoning trace" terminology. By intermediate tokens, we refer to the unfiltered tokens emitted by the LLM before the solution. This should be distinguished from post-facto explanations or rationalizations of the process or the product of said "thinking." For example, OpenAI o1 *hides* the intermediate tokens it produces (perhaps because they aren't that interpretable to begin with?) but sometimes provides a sanitized summary/rationalization instead. In contrast, DeepSeek R1 (DeepSeek-AI, 2025) provides the full intermediate token sequences (which often *run for pages* even for simple problems; see Figure 3). To be clear, our focus here is on the anthropomorphization of unfiltered intermediate tokens rather than such post-facto rationalizations. It is well known that for humans at least, such post-facto exercises are meant to teach or convince the listener, and may not shed much meaningful light on the processing that went in (Nisbett & Wilson, 1977).

We also consider any non-solution tokens corresponding to the external commitments made by the LLM in agentic scenarios (e.g. tool calls), or interventions from external sources (e.g. results of tool calls) as distinct from the "think traces," as these necessarily must have semantics outside of LLM; see Section 6.2. We should also clarify that our position and reservations are only about ascribing end user interpretability to intermediate tokens. This doesn't extend to efforts that attempt to analyze why and how intermediate tokens help the model itself (e.g. (Bogdan et al., 2025)).[2]

Finally, we note that our main argument is not about whether

---

[1]We will use the term *derivational trace* as a neutral stand-in for intermediate tokens, whether generated by humans, formal solvers or other systems, rather than the more popular anthropomorphized phrases "chains of thought" and "reasoning traces".

[2]In light of our own results (see Section 6.1), we speculate that the tokens provide a scaffold for the model to fit itself to the solutions.

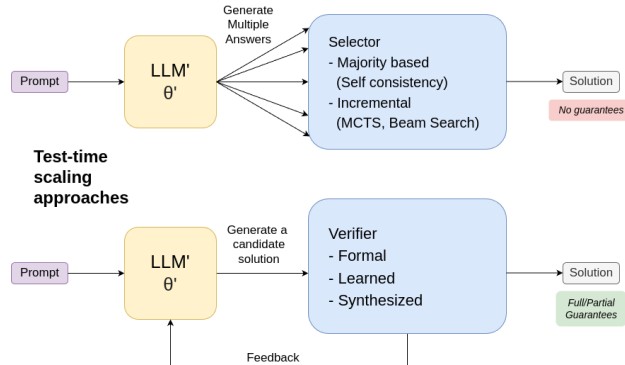

*Figure 1.* Test-time scaling approaches for teasing out reasoning

the LLM intermediate tokens exhibit "*human-like reasoning*"–which is clearly hard to pin down, but whether the reasoning can be said to lead to the solution in any logically interpretable sense. Specifically, whether the prompt plus intermediate tokens leads to the solution in some logical way (other than just changing the conditional distribution of the next token that LLMs anyway do). The works we survey provide clever ways of rigorously checking the logical validity of the trace leading to the solution–and find it lacking.

## 2. Background: Test Time Scaling, Post-Training & Derivitional Traces

In contrast to the pre-GPT4 LLMs pre-trained solely on our digital footprints, the so-called "reasoning models" that started with o1, (sometimes referred to as Large Reasoning Models or LRMs) have been built on insights from two broad but largely orthogonal classes of ideas (Kambhampati et al., 2025; Valmeekam et al., 2025): (i) **test-time scaling** techniques, which involve getting LLMs to do more work than simply guessing the most likely direct answer; and (ii) **post-training methods**, which involves shifting a version of test-time scaling to training phase, where the model is made to guess solutions to synthetic problem instances, and the trajectories ending in verified solutions are used to fine tune the model parameters.

### 2.1. Test-time Scaling

There is a rich history of approaches that use scalable online computation to improve upon faster initial guesses, including limited depth min-max, real-time A* search and dynamic programming, and Monte Carlo Tree Search (Russell & Norvig, 2010; Graves, 2016). Test-time scaling approaches (see Figure 1) mirror these ideas.

Perhaps the most popular and enduring class of test-time inference ideas involves generating many candidate solu-

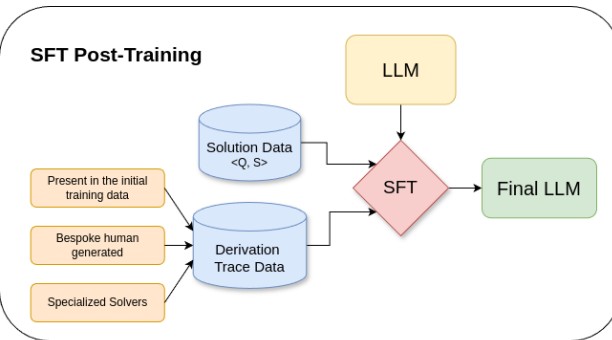

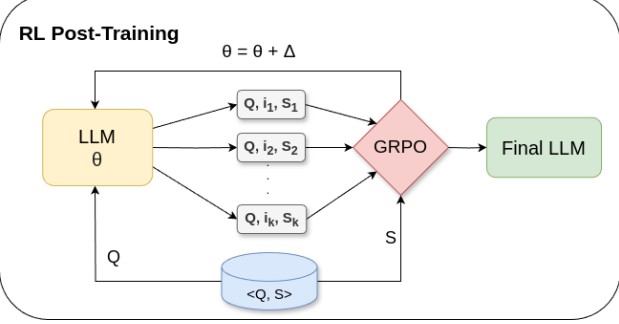

*Figure 2.* Post-training Approaches for teasing out reasoning

tions from an LLM and using some selection procedure to choose the final output. The simplest implementation is known as *self-consistency* (Wang et al., 2023b): choose the most common answer.

More sophisticated selection procedures, such as LLM-Modulo (Kambhampati et al., 2024), attempt to verify that an LLM's output is correct. When paired with an LLM in this manner, the combined system can be seen as a *generate-test* framework, and naturally raises questions about the verification process: *who does it*, and *with what guarantees?* A variety of approaches have been tried–including using LLMs themselves as verifiers(Yao et al., 2023) (although this is known to be problematic (Stechly et al., 2025)), learning verifiers(Arora & Kambhampati, 2023; Zhang et al., 2024), and using external sound verifiers that come with either full or partial guarantees. In cases where verifiers provide explanations or feedback when a guess is incorrect, these can be passed back to the LLM so it generates better subsequent guesses (Romera-Paredes et al., 2023; Trinh et al., 2024; DeepMind, 2025).

## 2.2. Post-Training and Intermediate Tokens

Unlike the test-time inference techniques, that augment the inference stage of standard LLMs, the post-training techniques are aimed at the LLM training stage. If we view the base model as a generator of plausible solutions to the reasoning problem, the test time scaling techniques implement a "generate test" paradigm, improving the ac-

curacy by checking the plausible solutions against a verifier. Post-training, in contrast, tries to shift the test part of this generate-test into the generator (model) itself[3] Unlike standard LLM fine tuning which uses ⟨ problem, solution⟩ pairs, (c.f. (Parthasarathy et al., 2024)) post-training uses ⟨ problem, intermediate tokens, solution ⟩ triples, and can be understood as compiling the signal from the verifier into the model parameters.

Using DeepSeek R1 (DeepSeek-AI, 2025) as a case study (see Figure 2), the model collects many synthetic problems, and for each generates plausible solution trajectories (comprising intermediate tokens followed by solution guesses). The solutions in these trajectories are evaluated by external problem-specific verifiers (DeepSeek calls them "rule-based reward models"). These trajectories with their rewards become the basis for a RL fine-tuning phase. The overall process has been termed RLVR–or RL with (externally) verified rewards (Gao et al., 2024; Lambert et al., 2024; Wang et al., 2025), but can be seen as shifting an LLM-Modulo like (Kambhampati et al., 2024) test-time scaling to training phase, and adding RL finetuning phase to update the model parameters.

## 2.3. Trace Generation for Training

A variety of approaches have tried to generate derivational traces to post-train LLMs, ranging from paying annotators for step-by-step derivations to generating and selecting them with LLMs. We classify these in terms of (i) how candidate traces are generated and filtered, and (ii) how they are used to improve the underlying LLM through supervised fine tuning or reinforcement learning; see Figure 2.

**Generating Candidate Derivational Traces:** Several trace generation methods were considered: (i) *Human-generated Traces:* An obvious way to obtain additional derivational data is to have humans create it (Lightman et al., 2023). (ii) *Solver-generated Traces:* Searchformer (Lehnert et al., 2024b), Stream of Search (Gandhi et al., 2024), as well as DeepMind's work in (Schultz et al., 2024; Markeeva et al., 2024) use a much more scalable approach by using standard search algorithms to produce datasets containing not just answers but also the execution traces generated along the way. (iii) *LLM-generated Traces:* Rather than creating high-quality traces from the start, an increasingly popular approach is to generate them from an LLM and filter afterwards (Kojima et al., 2022).

**Filtering Traces:** Naively LLM-generated traces are often not useful unless they are filtered. Researchers have varied in how they approach this trace selection process, ranging

---

[3]There is a famous dictum attributed to Marvin Minsky that *intelligence is shifting the test part of generate-test into generate part.*

from selecting only those that are correct at each step (according to human labelers), training process reward models that attempt to automate human verification(Lightman et al., 2023), to selecting traces by formally verifying whether they lead to correct final solutions without considering the trace content (Zelikman et al., 2022; DeepSeek-AI, 2025).

**Improving LLMs Using Derivational Traces:** Once derivational traces have been selected, they can be used to further train an LLM. Early approaches fine-tuned LLMs directly on such traces(Zelikman et al., 2022; Lehnert et al., 2024b; Gandhi et al., 2024). More recent advances, especially starting with DeepSeek R1, have been credited to the use of reinforcement learning. It has however been argued that the specific type of RL approach used in DeepSeek R1 is a close cousin of supervised fine tuning (Samineni et al., 2025).

## 3. Anthropomorphization of Intermediate Tokens

As we discussed, post-training can induce a model to first generate long strings of intermediate tokens before outputting its final answer. There has been a tendency in the field to view these intermediate tokens as the human-like "thoughts" of the model or to see them as *reasoning traces* which could reflect internal reasoning procedures. This is precisely the tendency our position paper argues against. We start by listing the various (unhealthy) ramifications of this anthropomorphization:

- Viewing intermediate tokens as reasoning/thinking traces has led to a drive to make them "interpretable" to humans in the loop (nevermind that interpretability mostly meant that the traces were in pseudo English). For example, DeepSeek (DeepSeek-AI, 2025) dabbled in training an RL-only model (R1-Zero) but released a final version (R1) that was trained with additional data and filtering steps specifically to reduce the model's default tendencies to produce intermediate token sequences that mix English and Chinese!

- It has led to an implicit assumption that correctness/interpretability of the intermediate tokens has a strong correlation, or even causal connection, with the solution produced. This tendency is so pronounced that a major vendor's study showing that LRM's answers *are not always faithful* to their intermediate tokens was greeted with surprise (Chen et al., 2025).

- Viewing intermediate tokens as traces of thinking/reasoning has naturally led to interpreting the *length* of the intermediate tokens as some sort of meaningful measure of problem difficulty/effort (Su et al., 2024; 2025), and techniques that increased the length

of intermediate tokens were celebrated as "learning to reason" (DeepSeek-AI, 2025). Simultaneously there were efforts to *shorten* intermediate traces produced and celebrate that as learning to reason efficiently (Arora & Zanette, 2025; Shrivastava et al., 2026).

- There have been attempts to cast intermediate tokens as learning some "algorithm" that generated the training data. For example, the authors of Searchformer (Lehnert et al., 2024a) claim that their transformer learns to become "more optimal" than A* because it produces shorter intermediate token traces than A*'s derivational trace on the same problem.[4]

These corollaries, in turn, have lead to research efforts, which, when viewed under the lens of our position, become questionable enterprises (as we shall discuss in the following sections).

## 4. On the Questionable Semantic Status and Interpretability of Intermediate Tokens

### 4.1. Intermediate Tokens and End User Interpretability

While the fact that use of intermediate tokens during training and inference stages seem to improve LLM performance is beyond dispute, there are significant questions on whether these traces have any valid semantic import to the end user. The fact that intermediate token sequences often reasonably look like better-formatted and spelled human scratch work – mumbling everything from *"hmm..."*, *"aha!"*, *"wait a minute"* to "interesting." along the way – doesn't tell us much about whether they are used for anywhere near the same purposes that humans use them for, let alone about whether they can be used as an interpretable window into what the LLM is "thinking".

Famously, DeepSeek's R1 paper claimed that one of the most impressive observed behaviors of their trained models was the so-called "aha" moment: as part of the chain of thought it was producing in order to answer some question, the model output the token "aha", seeming to indicate that it had come upon a sudden realization. While a human may say "aha" to indicate exactly a sudden internal state change, this interpretation is unwarranted for models which do not have any such internal state, and which on the next forward pass will only differ from the pre-aha pass by the inclusion of that single token in their context. Interpreting the "aha" moment as meaningful exemplifies the long-neglected assumption about long CoT models – the false idea that derivational traces are semantically meaningful, either in resemblance to algorithm traces or to human reasoning. Fur-

---

[4]Taken literally, this would be a rather questionable claim, considering that A* search is a provably optimal algorithm for graph search (Hart et al., 1968).

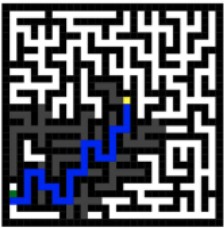

*Figure 3.* Intermediate tokens DeepSeek-R1 produces for a planning problem.

ther, there have also been works which attribute cognitive behaviors (like backtracking, self-verification etc.) to the models based on their reasoning traces and try to induce these kinds of behaviors through examples in the hope of improving the models' performance (Gandhi et al., 2025; Qin et al., 2025).

One reason that this anthropomorphization continues unabated is because it is hard to either prove or disprove the correctness of these generated traces. DeepSeek's R1, even on very small and simple problems, will babble pages and pages of text in response to each and every query (a snippet is shown in Figure 3), and it is far from clear how to verify if these monologues constitute sound reasoning. Arguably, people examine the traces in places, see familiar phrases reminiscent of what a human solving such a problem might utter, and assume that the reasoning trace sounds plausible. While there have been some valiant efforts to make sense of these large-scale mumblings–e.g. (Marjanović et al., 2025)– the analyses here tend to be qualitative and suggestible reminiscent of "lines of code" analyses in software engineering. It is no wonder then that few if any LRM evaluations even try to check their pre-answer traces, and focus only on evaluating the correctness of their final answers.[5]

**Studies on Training Transformers with A\* Search Traces on Mazes:** While evaluating the intermediate tokens produced by general LRMs may be out of direct reach, the traces generated by format-constrained models trained to imitate the derivational traces of domain-specific solvers can be formally verified. In (Valmeekam et al., 2026) we report on a series of experiments with transformers trained on a

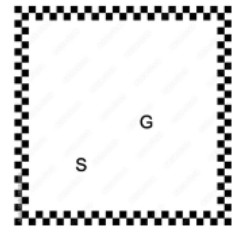

*Figure 4.* On the left is an example of maze path finding instances used to train the transformer in (Valmeekam et al., 2026). On the right is a trivial "no-maze" maze instance used to show the disconnect between intermediate token length and problem complexity.

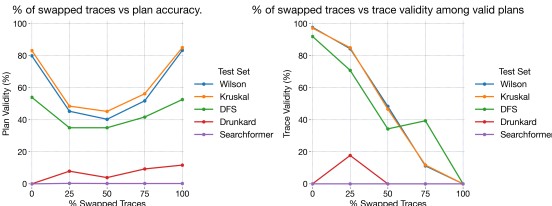

*Figure 5.* A study reproduced from (Valmeekam et al., 2026) showing the curious phenomenon that as the models are trained with increasingly incorrect–in their case "swapped" traces–the inference time accuracy of the resulting model is high both with fully correct and fully swapped traces, dipping only in the middle.

corpus of maze path finding instances. The maze instances– such as the one shown on the left in Figure 4–are generated with a variety of distributions. They are solved with A\* search (Hart et al., 1968), and the problem instance, the A\* search trace (of open and closed list manipulations), and the path found by the A\* search are used to train the transformer. Since the traces are generated by A\* search, at inference time, the validity of A\* search trace-like tokens produced by the model can be verified by A\* to see if they indeed lead to the solution produced. Our findings show that there is only a loose correlation between the validity of the trace and the correctness of the solution plan, especially when the problem instances go out of the training distribution.

We then report a causal intervention, training additional models on noisy or irrelevant traces and find that there are (nonsensical) trace formats–involving derivational traces *swapped* between instances–that nevertheless maintain or even increase the model's performance. Our experiments with models trained on a mix of instances with correct vs. swapped traces (see Figure 5) suggest that the test time solution accuracy remains high both with fully correct and fully swapped traces, dipping only when the traces are mixed. This suggests that what matters for a model to improve accuracy with intermediate tokens is not their semantic import, but perhaps a consistent pattern in the training data for the model to fit itself.

---

[5]Approaches like Process Reward Models (Zhang et al., 2025b) try to make the reasoning traces a bit more locally consistent–but have taken a back seat since the success of DeepSeek R1.

**Effects of RL Post-training on Trace Validity:** If the intermediate tokens produced by models that are explicitly trained on correct traces are still not guaranteed to be valid during inference time, then there seems to be little reason to believe that trace validity improves when these models are further post-trained with outcome-based RL. This is because such post-training techniques (DeepSeek-AI, 2025; Samineni et al., 2025) change the base model parameters to bias it more towards the trajectories that end up on solutions verified correct by the external verifiers during training. Indeed in (Valmeekam et al., 2026) we examined the effects of post-training with RL, specifically GRPO, on semantic correctness of reasoning traces. We report that post-training doesn't necessarily increase the semantic validity of the traces even though it improves the solution accuracy. We find that even in models trained on irrelevant (swapped) traces, post-training improves solution accuracy even though the trace validity remains very low. This should not be surprising given that most works that do these types of post-training reward only the solution accuracy and ignore the content of intermediate tokens (DeepSeek-AI, 2025; Yu et al., 2025).

**Studies on Pretrained models in QA Tasks:** We have also conducted an investigation to test the correlation between intermediate traces and final solution performance with pretrained models from Llama and Qwen families fine tuned in the Question-Answering (QA) domains (Bhambri et al., 2026). By decomposing the QA reasoning problems into verifiable sub-problems that can be evaluated at inference time, we first generated a Supervised Fine-Tuning (SFT) dataset with correct intermediate traces paired with correct final solutions. To carry out an intervention experiment, we generated another SFT dataset consisting of incorrect intermediate traces again paired with correct final solutions. For the first SFT experiment setting, the results show a large number of false positives where the fine-tuned models output correct final solutions but incorrect intermediate traces. Interestingly, the intervention experiments with incorrect intermediate traces even outperforms the SFT with correct intermediate trace setting. We also show empirically that trace correctness does not guarantee final solution correctness. Similarly, final solution correctness also does not imply that they were preceded by semantically correct intermediate traces.

In a related study comparing the correlation between end-user interpretability and SFT performance, we (Bhambri et al., 2025; 2026) showed via systematic human subject studies that there is a mismatch between the derivational traces that help the underlying model in terms of solution accuracy, and those that help end users. Specifically, the long and meandering derivational traces produced by the DeepSeek R1 model lead to better solution accuracy in the distilled model, yet score lowest on dimensions of user pref-

erence such as interpretability, faithfulness, and predictability, as well as their utility to the end-users as measured by the comprehensibility of these traces. In contrast, the verifiable traces that users found most comprehensible, interpretable and least cognitively demanding did not yield comparable solution accuracy. This suggests a strong decoupling, as the traces that are least interpretable to users and are hard for them to parse are the ones that most benefit the model.

**Other Studies:** Other studies on training with noisy traces support these conclusions, with findings that show how LLMs remain robust to semantic noise in traces and similar performance gains can be achieved without semantic correctness (Li et al., 2025a; Su et al., 2024). Li et al. (Li et al., 2025a) perform model distillation using noisy traces on math and coding problems and find that the smaller LLM that is being trained remains largely robust to the semantic noise in the trace. Even when trained on derivational trace containing largely incorrect mathematical operations, the LLM shows significant performance improvements as compared to the base model. Dualformer (Su et al., 2024), an extension of Searchformer (Lehnert et al., 2024a), which trains transformer models on truncated A* derivational traces (by arbitrarily removing steps from the original A* search process–and thus destroying any trace semantics) to improve solution accuracy, is another evidence for performance improvements with wrong traces! Other works that demonstrate how reasoning traces are not reliable indicators of the model's internal computations include (Baker et al., 2025; Korbak et al., 2025b; Chua et al., 2025; Greenblatt et al., 2024; Chen et al., 2025; Arcuschin et al., 2025)

## 4.2. Intermediate Token Length and Problem Complexity

The length of the intermediate tokens have also been subject to anthropomorphization as indicative of "thinking effort." DeepSeek R1 (DeepSeek-AI, 2025), claims that RL post-training is *learning to reason* as shown by the increased length of intermediate tokens over RL epochs. Since the vendors started charging end users for the intermediate tokens despite not showing them (Sun et al., 2025; Valmeekam et al., 2025), ironically there have been subsequent efforts to *reign in* the intermediate token lengths, and claim that as a way to reduce compute while preserving task performance/accuracy (c.f. (Arora & Zanette, 2025; Shrivastava et al., 2026)). On a related front, researchers from cognitive science have also tried to draw correlations between human thinking cost and LLM thinking cost as measured in intermedate token lengths (de Varda et al., 2025), receiving some push back (Hu, 2026; Vankov et al., 2026).

In (Palod et al., 2025; Valmeekam et al., 2026), we also examined the trace lengths of models trained on A* search traces on problems of varying difficulties. We found that al-

though trace lengths can look indicative of problem adaptive computation when tested on in-distribution problems, this correlation breaks down when the problem instances are out-of-distribution. In one of our experiments (see Figure 4), we found that a transformer model trained on complex mazes (such as the one shown on the left in Figure 4) fails on trivial "no-maze" instances (with start and goal points entirely in free space, as shown on the right in Figure 4). These instances would require minimal computation for A* search, and yet the transformer model often produces extremely long derivation traces, in many cases even exhausting the context window. These findings indicate that the correlation is quite tenuous between the from-scratch computational complexity of the problem and the derivational trace produced by the LLM.

The original DeepSeek R1 argument (Guo et al., 2025) interpreting increased intermediate token length as indicative of improved thinking abilities has been called into question by subsequent work (Samineni et al., 2025; Fatemi et al., 2025). In (Samineni et al., 2025) we examine the MDP formulation used in DeepSeek R1 and show that with the structural assumption of representing states as sequences of tokens, and uniformly distributing the terminal reward into intermediate tokens, RL is incentivized to generate longer intermediate token sequences–something that has been misattributed to "improved reasoning." At some level, this should not be surprising given that the whole point of RL is to figure out credit assignment, and the division of final reward equally into intermediate tokens short circuits this process, making RL close to a filtered on-policy version of supervised fine tuning (Samineni et al., 2025). In (Kalwar et al., 2026; Samineni et al., 2025), we also argue that works promising "efficient reasoning" by reducing the length of interemediate tokens, such as (Arora & Zanette, 2025; Shrivastava et al., 2026) are better understood as over training the base model for a particular distribution of problem instances.

### 4.3. Intermediate Tokens and False Trust

The expectation that intermediate tokens can shed meaningful light on the internal operation of the LLM in arriving at the answer has also lead to proposals to use them to modulate end user's trust in the solution. Indeed, a significant body of work has focused on improving the faithfulness of the intermediate tokens, treating them as user-facing explanations that reflect the model's reasoning process and how it arrived to the answer (Li et al., 2025b; Tanneru et al., 2024; Paul et al., 2024; Wei Jie et al., 2024). Frontier models have similarly framed displaying the intermediate tokens to end-users as a feature that improves the transparency of the reasoning models to these users (Guo et al., 2025). However, since the intermediate tokens often imitate the style of human chains of thought, treating them as explanations might wind up engendering false trust in the end users.

In recent work (Palod et al., 2026), we directly evaluate this risk through a human subject study examining the effect of showing intermediate tokens or their summaries alongside the final answer, in the realistic setting where end-users cannot independently verify the solution. The results show that reasoning traces and their summaries increase user trust in the model's predictions regardless of their correctness, thus substantially increasing false trust of the user (users become more likely to accept incorrect answers than when shown the answer alone).

## 5. Alternative Views

We have made it clear from the outset that there certainly are alternate views about the semantic status of the intermediate tokens–indeed their prevalence and popularity is the main reason motivating this position paper. To summarize, the phrase "chains of thought" originally arose as a way of prompting LLMs to elicit particular types of prompt completions ("behaviors") (Wei et al., 2022; Kojima et al., 2022). Originally such CoT's were meant to be hand-crafted by the end users and include human interpretable advice that the LLMs were seen to be following. Later studies, such as (Stechly et al., 2024; Wang et al., 2023a) pushed back on the alignment between the advice and the completions.

With the advent of reasoning models such as DeepSeek R1, the CoT terminology has been repurposed to refer to the intermediate tokens that the models are trained to produce on their way to the solutions. These tokens have been analyzed for potentially human interpretable patterns. The DeepSeek R1 paper itself (DeepSeek-AI, 2025) helped this narrative along by analyzing the intermediate tokens for the presence of phrases that, when used by humans, typically suggest reflection and insight. In their paper, they talk about the *aha* moment in R1's intermediate tokens. Latter work such as *thoughtology* (Marjanović et al., 2025) took this narrative further by looking for correlations between specific types of passages in the intermediate tokens (as extracted *post-facto* by another LLM) and the solution accuracy. More recently, another group (Kim et al., 2026) extended the same type of LLM-based analysis of the intermediate tokens generated by a reasoning model–this time in terms of shifting voices/perspectives–and claimed that *reasoning models generate societies of thought*, and implied that this is what explains their effectiveness. It should be noted that these analyses are often qualitative, and fail to establish direct connection between the narrative of the intermediate tokens and the final result. In Section 4.2, we also mentioned and critiqued works that equate the length of intermediate tokens with thinking effort.

Given that the current models are trained on large corpora of human data, the fact that they produce intermediate tokens ("chains of thought") that sound plausibly like those that

might be generated by humans may well be a form of imitating *cultural routines* (c.f. (Gopnik, 2016)) in the training data. Thus, our position is not that intermediate tokens will never have passages that might be interpretable by humans as corresponding to reasoning, but that such interpretability may be accidental and cannot be relied upon by the end users to assess their trust in the solutions provided by the models. Even such accidental interpretability might dissipate as models are increasingly post-trained with outcome reward-based RL (DeepSeek-AI, 2025). Interestingly, some works such as (MacDiarmid et al., 2025) characterize this lack of connection between intermediate tokens and final solution as indication of models learning to cheat!

A related issue is that none of the major frontier model makers–OpenAI, Google, Anthropic–show their actual intermediate tokens for citing proprietary concerns (although they do continue to bill the end user for that, raising auditing concerns (c.f. (Sun et al., 2025)). The model card for GPT-OSS (OpenAI et al., 2025), the open-weight reasoning models released by OpenAI, states that they use Harmony Response Format, which has three channels, *analysis, commentary* and *final*. The *analysis* part seems to correspond to the intermediate tokens (that are not shown in their production models), and the *final* part corresponds to the solution tokens. The *commentary* part typically has high level commentary interpretable for the end user, and is admittedly distinct from the *analysis* part that corresponds to intermediate tokens. It is not clear how and when the *commentary* part is generated. It is clear that their production models only show the summary part, and not the actual intermediate tokens that are the subject of post-training.

Ironically, the increasing realization that intermediate tokens may not have interpretable semantics has lead some researchers to issue public entreaties to the frontier model makers to preserve some semblance of interpretability in CoTs so the models can be monitored (Korbak et al., 2025a).

# 6. Implications and Call to Action

While some anthropomorphization can be harmless metaphors, we argued that viewing intermediate tokens as reasoning traces or "thinking" is actively harmful, because it engenders false trust and capability in these systems, and prevents researchers from understanding or improving how they actually work.

To the extent the research community finds our position persuasive, our recommendation is to stop assuming (or looking for) end user semantics in the intermediate tokens produced by the reasoning models. Human interpretation of intermediate tokens should not be used as a proxy measure for the trustworthiness of the solutions.

Given that the intermediate tokens may not have any seman-

tic import, deliberately making them *appear* more human-like is dangerous. In the end, LRMs are supposed to provide solutions that users don't already know (and which they may not even be capable of directly verifying). Engendering false confidence and trust by generating stylistically plausible ersatz reasoning traces seems ill-advised! After all, the last thing we want to do is to design powerful AI systems that potentially exploit the cognitive flaws of users to convince them of the validity of incorrect answers.

Where trust in the final solution is needed, it should instead come from verification of the correctness of the solution itself by the end users or third party sources–including problem class specific verifiers (c.f. (Kambhampati et al., 2024)).

Given that intermediate tokens are meant mostly to help LLMs, restricting them to some superficial linguistic format with hopes that it will be more palatable to end users becomes quite an albatross. This is a lesson from DeepSeek R1 (DeepSeek-AI, 2025) that is often missed. When they re-trained their original R1-Zero model–that happened to produce a combination of English and Chinese tokens, with a costly supervised fine tuning phase on carefully curated English intermediate tokens generated by humans, the performance (as measured in solution accuracy) worsened, without any concomitant measured improvements in the actual validity of the intermediate tokens generated!

Once we stop ascribing questionable interpretability to the intermediate tokens, and recognize that they are meant to help the LLM and not the end user, that would also free us to train models that optimize the intermediate tokens only for solution accuracy–even if the intermediate tokens themselves don't any longer look like plausible language utterances that humans might exhibit. This could, in theory, allow models to consider intermediate tokens made up of non-linguistic tokens–basically any vector from the embedding space, even if it doesn't correspond to a unique vocabulary item. Already there is some evidence that such methods can lead to further improvements in solution accuracy (c.f. (Hao et al., 2024; Zhang et al., 2025a)).

## 6.1. Viewing Intermediate Tokens as Learned Prompt Augmentations

In terms of explaining the role of intermediate tokens in the performance of current reasoning models, our main contribution is to urge the community to look beyond semantics of the reasoning traces, or their correlation to problem complexity. One speculative future direction that we believe merits investigation is viewing intermediate tokens as prompt augmentations (Kambhampati et al., 2025). The intuition is that for a given task prompt $T$, there may exist an augmentation PA which boosts the LLM's performance on that task:

$$\mathbb{P}\big(\mathrm{Sol}\big(\mathrm{LLM}(T + \mathrm{PA}),\, T\big)\big) > \mathbb{P}\big(\mathrm{Sol}(\mathrm{LLM}(T),\, T)\big)$$

Here $\text{Sol}(y, T)$ indicates that $y$ solves $T$, and $\text{LLM}(x)$ is the model's completion for input $x$. The central challenge then is to learn the Skolem function

$$\text{PA} = f_\theta(T, \text{LLM}),$$

that maps each task to an effective augmentation. This can be accomplished through modifying the model itself to inherently and automatically augment prompts, as is the case in models that first generate long chains of intermediate tokens before their final answers. Crucially, prompt augmentations have no need to be human-interpretable. In fact, we see results that back this up in the adversarial prompting literature, where effective jailbreaks can be effected by augmenting prompts with human-uninterpretable strings (Zou et al., 2023; Cherepanova & Zou, 2024; Liu et al., 2024; Hackett et al., 2025) or modifying them with random syntactic permutations, capitalizations, and shufflings (Hughes et al., 2024), as well as the recent work on using intermediate tokens from the continuous latent space (Hao et al., 2024). In this prompt augmentation view, intermediate tokens (and their length) can perhaps be interpreted as the scaffold that the reasoning model uses to learn to manipulate it's context to bring the current instance closer to its training distribution, rather than necessarily a reflection of the computational hardness of the problem instance.

### 6.2. Agentic Systems and External vs. Internal Intermediate Tokens

We have focused on the intermediate tokens produced by standard reasoning models (such as DeepSeek R1) that just produce the so-called chains of thought followed by their solution guess. In these situations, only the solution tokens need to make sense to the end users (or the external programs such as verifiers). Things change when we use LLMs in the so-called *agentic* scenarios, where before outputting the final solution, the model might send out tool calls, invoking the external tools, and incorporating the tool results into the context window. In such cases, the context window consists of at least three types of tokens interleaved multiple times: (i) internal "chain-of-thought" tokens produced by the model (ii) tool calls generated by the model and (iii) results of the tool calls from the external tools. Although the literature unfortunately uses the catch-all term "thinking traces" to refer to all these intermediate tokens,[6] they have quite different properties. In particular, only the first type correspond to the intermediate tokens discussed in this paper that don't need to have any semantics outside of the LLM. The second type–tool calls, correspond to communications with external tools and should make sense to them.

---

[6]For example, a recent paper (Ríos-García et al., 2026) arguing that AI Scientists arrive at their answers without being faithful to their reasoning traces, actually refers to the tool calls and associated results as the reasoning traces

Crucially these tool call tokens correspond to *commitments* by the LLM to the external environment, and may not necessarily be reversible if the environments are *non-ergodic*. In (Bhat et al., 2026), we show how such tool calls can be viewed as intermediate steps taken by the LLM towards a solution, and how a generalization of LLM-Modulo, called *LLM-Process-Modulo*, can be used to control the run time behavior and efficiency of LLMs.

## 7. Conclusion

In this position paper, we argued against the popular tendency in the LLM research community to anthropomorphize intermediate tokens as reasoning or "thinking". We note that our position is not that intermediate tokens can't ever have interpretable meaning but that they don't necessarily have to have any interpretable meaning. Any human interpretable meaning in the intermediate tokens might be a fortuitous coincidence of such rationales being present in the training data, rather than the model actually doing internal computations corresponding to those statements. In other words, the correlation here may be spurious rather than causal. This seems to be the clearest lesson to be drawn from the cited recent literature.

Our position is bolstered by the facts that (1) there certainly isn't any theory drawing causal connections between the semantics of the intermediate tokens and the final solution– beyond the basic understanding that intermediate tokens change the conditional distribution of the solution tokens and (2) even the proponents of user interpretable semantics for intermediate tokens seem to realize this and write papers talking about why it is incumbent to "protect" the "fragile" connection between intermediate tokens and final solutions so as to allow for some kind of "monitorability" of LLMs (even if it may well be only illusory) (Korbak et al., 2025b).

Until there is evidence—beyond circumstantial—-that LRM reasoning tokens correspond to internal computations, it seems unwise to depend on the intermediate tokens for "safety monitoring" in safety-critical domains. A third-party verification of the solutions/decisions seem to be the better way to go in such scenarios.

As we have mentioned in Section 5, most frontier models, with the exception of DeepSeek R1, already seem to, in effect, abide by our position in that they are no longer showing the intermediate tokens anyways (citing proprietary considerations). Ironically it is the research community that still seems to entertain the possibility that intermediate tokens can provide an interpretable explanation of the model's operation to the end user. This paper is thus a modest attempt to persuade the community away from such anthropomorphization.

## Acknowledgments

This research is supported in part by grants from DARPA (HR00112520016), ONR (N00014-25-1-2301 and N00014-23-1-2409), DoD RAI (via CMU subcontract 25-00306-SUB-000), an Amazon Research Award, and a generous gift from Qualcomm. We thank Atharva Gundawar, who was part of many early discussions. We also thank Tom Dietterich for thoughtful feedback on many of these ideas during his visits to ASU.

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
