# OpenReview forum: "Position: Stop Anthropomorphizing Intermediate Tokens as Reasoning/Thinking Traces!"
_ICML.cc/2026/Position_Paper_Track — ICML 2026 Position Paper Track regular_

### Official Review · Reviewer_wdHZ · 2026-03-04

**Significance:** 3
**Argument Clarity:** 1
**Rating:** 3
**Confidence:** 2

**Questions:**

- I think one important thing for more discussions on this topic is that, if we stop to think about all the human intuitions about the reasoning traces, how should we treat them? Simply as internal representations of neural models, like other hidden vectors?

**Alternative Views Section:**

Yes

**Compliance With Llm Reviewing Policy A Conservative:**

Affirmed.

**Discussion Potential:**

1

**Final Justification:**

The score is raised since some of my concerns are addressed after the rebuttal.

**Paper Summary:**

This work argues against the view that the intermediate tokens can be regarded as human-like reasoning traces. This work raises some important questions on whether these traces have any valid semantic for the end user and presents some evidences that such anthropomorphization may confuse the nature of the underlying models.

**Position:**

Yes

**Position In Title:**

Yes

**Related Work:**

2

**Strengths And Weaknesses:**

Strengths:
- The topic about how to interp reasoning traces is relevant and important to the field of LLMs.
- This work provides some evidences that it is difficult to understand and interpret the reasoning tokens.

Weaknesses:
- My main concern is that anthropomorphization itself, the main target this work is arguing against, seems very fuzzy and difficult to have a clear definition. I think it is more about using human intuition to motivate the design of the reasoning models, rather than some more detailed approaches. In this case, I think there should clear definitions should be established first, otherwise the follow-up dicussions might be confusing.
- From the aspect of presentation, this work can be further improved. For example, Section 4 is relatively difficult to read. It would be helpful if it can be split into sub-sections or there can be annotations added to make the discussions more structured.

**Support:**

2

---

> ### Author Rebuttal · Authors · 2026-03-31
>
> Thank you for your comments. Here are our responses. We hope they, along with the comments from the fellow reviewers, persuade you to reconsider your evaluation.
>
> >I think it is more about using human intuition to motivate the design of the reasoning models, rather than some more detailed approaches
>
> We respectfully beg to differ on this. As we discuss in Section 3, multiple papers and authors have explicitly advocated viewing intermediate tokens as “reasoning traces” that help the end user understand the internal operations of the LLM. It is this anthropomorphization–rather than just the use of anthropomorphic terms–that we advocate against.
>
> To elaborate, while we prefer the neutral term “intermediate tokens” to refer to the tokens LLMs output before the solution, we should point out that our position is not against the “terminology” (however anthropomorphic), but against the subsequent anthropomorphization.  Many anthropomorphic terms in LLM literature, such as “thoughts”, “CoTs” have become too entrenched at this point to try to replace them. Our position is to caution the community that the anthropomorphization of actually  assuming that the  intermediate tokens must be telling the end user about LLM intermediate reasoning steps just because we **called them reasoning traces.**
>
> > If we stop to think about all the human intuitions about the reasoning traces, how should we treat them? Simply as internal representations of neural models, like other hidden vectors?
>
> Our position is not that intermediate tokens can't ever have interpretable meaning  but that they don't necessarily have to have  any interpretable  meaning.  The intermediate tokens are best seen as “prompt augmentations” that help the LLM.
>
> Any  human interpretable meaning in the intermediate tokens might be a  fortuitous coincidence of such rationales being present in the training data, rather than the model actually doing internal computations corresponding to those statements. In other words, the correlation here may be  spurious rather than causal. This seems to be the clearest lesson to be drawn from the cited recent literature.
>
> Our position is bolstered by the facts that (1) there certainly isn't any theory drawing causal connections between the semantics of the intermediate tokens and the final solution--beyond the usual "intermediate tokens change the conditional distribution of the solution tokens" and (2) even the proponents of user interpretable semantics for intermediate tokens (c.f. Korbak et. al.)  seem to realize this and write papers talking about why it is incumbent to "protect" the "fragile" connection between intermediate tokens and final solutions so as to allow for some kind of "monitorability" of LLMs (even if it may well be only illusory).
>
> Given the fragility of the intermediate tokens, as well as the fact that frontier models don't actually show reasoning tokens--but only separately produced summaries--this paper cautions against uncritically assuming that the tokens have end user semantics.
>
> Until there is evidence–beyond circumstantial–that LRM reasoning tokens correspond to internal computations, it seems unwise to depend on the intermediate tokens for “safety monitoring” in safety-critical domains. A third party verification of the solutions/decisions seem to be the better way to go in such scenarios.
>
> At the least, we believe that our position should be aired as a countervailing view until someone actually shows the conditions under which we can ensure causal connection between intermediate token semantics and the LRM solutions--beyond mere change of conditional distribution--which can also be got without using the tokens that have meaning for end users (such as latent tokens).
>
> >From the aspect of presentation, this work can be further improved. For example, Section 4 is relatively difficult to read. It would be helpful if it can be split into sub-sections or there can be annotations added to make the discussions more structured.
>
> We will certainly take your suggestions about reorganization of Section 4 into subsections for the revised version. We also note that in addition to the cited literature, our position is also motivated by some of the supporting studies we have done. Once the double-blind restriction is lifted for the final version, we will clearly point out which works were done by our group vs. other groups, and we believe that will make the readers realize that our position is certainly motivated in part by our own direct studies.

---

> > ### Author Rebuttal · Reviewer_wdHZ · 2026-04-02
> >
> > Thank you for your response. One specific interesting point to investigate would be the inconsistency between the intermediate tokens (or "reasoning traces" as called by many) and the final answer. For example, sometimes we may have wrong process but correct answer. It might be helpful if more related evidence (especially with recent models) can be provided.

---

### Official Review · Reviewer_42tp · 2026-03-09

**Significance:** 4
**Argument Clarity:** 2
**Rating:** 4
**Confidence:** 4

**Questions:**

Please refer to the weakness. I think the paper is a meaningful position paper but need further polishing.

**Alternative Views Section:**

Yes

**Compliance With Llm Reviewing Policy A Conservative:**

Affirmed.

**Discussion Potential:**

2

**Final Justification:**

After the rebuttal, my concerns are partially solved. My concern remains in "what we can do and what we need to do after understanding the position". But i decided to raise the score to 4.

**Paper Summary:**

The paper discusses intermediate tokens, which are often considered reasoning traces in large reasoning models. The authors first propose that these intermediate tokens may not reflect real “reasoning” in the same way that reasoning occurs in the human brain. They support this position from several aspects. First, they examine common markers in reasoning traces, such as “aha,” and argue that these markers do not necessarily indicate genuine reasoning, but may instead be overfitted patterns that appear in long intermediate outputs.

They then cite experimental results from related work showing that even when models are trained with incorrect intermediate tokens, their reasoning performance can still remain strong. This suggests that the intermediate tokens themselves may not semantically contribute to the correctness of the final answer. Furthermore, they argue that interpreting longer intermediate token sequences as evidence of stronger reasoning ability is not necessarily justified.

Finally, the authors conclude that researchers should be cautious about treating intermediate tokens as human-like reasoning processes and should reconsider the practice of continually increasing their length.

**Position:**

Yes

**Position In Title:**

Yes

**Related Work:**

2

**Strengths And Weaknesses:**

Strengths:

First, the paper selects a very meaningful topic and clearly states its position. The authors share the intuitive feeling that LLMs may not reason in the same way as humans, even after being trained with RL algorithms such as GRPO. This perspective may encourage researchers to rethink the role of reasoning traces and avoid training LLMs with increasingly long intermediate steps. In addition, the paper is closely related to several active research directions, such as latent reasoning and token auditing.

The paper supports its position through intuitive, experimental, and theoretical arguments. Although many of the claims are primarily intuitive (which is understandable for a position paper), the main argument is reasonably supported. The paper may encourage researchers to conduct more detailed and insightful studies on this topic, which currently appears to be underexplored.

Weaknesses:

As a position paper, my main concern is that the paper does not provide sufficient guidance for researchers who plan to work on this topic. The paper mainly claims that intermediate tokens do not represent human-like reasoning processes, but it does not offer guidance (even intuitive guidance) on several important questions:
(1) What constitutes human-like reasoning, and how could it be reflected in LLMs? How can we establish or evaluate human-like reasoning in LLMs?
(2) How can researchers rigorously determine whether intermediate tokens do or do not correspond to real reasoning?
(3) What kinds of research directions should be pursued regarding intermediate tokens?

Without addressing these questions, the paper reads more like an intuitive position statement rather than a perspective that provides actionable guidance for future research.

The structure of Sections 3 and 4 is also not well organized. The arguments supporting the paper’s position are not clearly structured and sometimes appear scattered. As a result, it is difficult for readers to follow the main line of reasoning and understand the overall outline. One possible improvement would be to organize the supporting evidence into several categories, such as: intuitive analysis (e.g., the discussion of “aha”), empirical evidence (e.g., Figure 4), and other observations (e.g., claims that some papers overemphasize reasoning length). Currently, these different types of arguments are somewhat entangled.

The writing of the paper could also benefit from further polishing. Some figures contain very small text that is difficult to read. In addition, some sentences appear overly AI-generated. For example, similar concepts are sometimes expressed using inconsistent wording, which makes the text harder to follow. Footnote 5 also seems unusual in formatting (though I may be mistaken if this is the intended format).

Some related fields may be value to discuss to support the position, e.g., latent reasoning[1] and token inflation[2].

[1]Hao, S., Sukhbaatar, S., Su, D., Li, X., Hu, Z., Weston, J., & Tian, Y. (2024). Training large language models to reason in a continuous latent space. arXiv preprint arXiv:2412.06769.

[2]Sun, G., Wang, Z., Zhao, X., Tian, B., Shen, Z., He, Y., ... & Li, A. (2025). Invisible tokens, visible bills: The urgent need to audit hidden operations in opaque llm services. arXiv preprint arXiv:2505.18471.

**Support:**

3

---

> ### Author Rebuttal · Authors · 2026-03-30
>
> Thank you for your close reading, thoughtful comments and your overall support of our position.
> Here are specific responses to your comments
>
>
> **What constitutes human like reasoning:** We would like to clarify that our main argument is not about whether the LLM intermediate tokens exhibit  "human-like reasoning"--which is clearly hard to pin down, but whether the reasoning can be said to lead to the solution in any logically interpretable sense..  Specifically, whether Prompt + reasoning trace leads to the solution in some logical way (other than just changing the conditional distribution of the next token that LLMs anyway do). The works we cite--such as Valmeekam et. al. 2025, and Bhambri et al. 2025a--provide clever ways of rigorously checking the logical validity of the trace leading to the solution--and find it lacking.
>
> Our position is bolstered by the facts that (1) there certainly isn't any theory drawing causal connections between the semantics of the intermediate tokens and the final solution--beyond the usual "intermediate tokens change the conditional distribution of the solution tokens" and (2) even the proponents of user interpretable semantics for intermediate tokens (c.f. Korbak et. al.)  seem to realize this and write papers talking about why it is incumbent to "protect" the "fragile" connection between intermediate tokens and final solutions so as to allow for some kind of "monitorability" of LLMs (even if it may well be only illusory).
>
> **What kinds of research directions should be pursued:** Section 6 (page 7) of our paper contains a clear call to action that our position entails. In particular, we point out that once the research community recognizes that the intermediate tokens may not have any semantic import, it both avoids the trap of engendering false trust in the end users, and also opens up research directions that will allow LRMs to be trained with intermediate tokens that don’t need to have any “human-like” syntax.
>
> In other words, we recommend that intermediate tokens be viewed just as prompt augmentations that the LRM learns during training to increase the conditional probability that the prompt completion contains a valid solution to the reasoning problem in the prompt.  In this context, we point out directions like using continuous latent space tokens in the LRM traces.
>
> We will make sure that these actionable directions are made more prominent in the revised version.
>
> **Related works may be valuable to support the position:** Thank you for the citations to Hao et. al. and Sun et. al. We note that we already discuss the connection to latent space tokens, and cite Hao et. al. in Section 6 (page 8, lines 405-418, left column).  Thanks for bringing  Sun et. al.’s paper on invisible tokens and visible bills to our attention; we agree that it is very much consistent with our position and we will cite it in the revised version. We do note that the general point they make is somewhat covered by our discussion in Section 4.1 (Intermediate token production and Problem adaptive computation).
>
>
> **Reorganizing Section 4.1:**  We will certainly consider your helpful comments for reorganizing Section 4 in our revised version.
>
>
> **Sentences appear overly AI-generated:** We will look at your formatting and rewriting suggestions carefully. We do want to emphasize that all of the writing in this paper was done by hand (and most of it by the lead author--who is allergic to AI Slop and thinks--possibly mistakenly--that LLMs don't write as well as them).

---

> > ### Author Rebuttal · Reviewer_42tp · 2026-04-03
> >
> > Thank you for the detailed rebuttal. While we appreciate the clarifications and additional context, we remain unconvinced that the paper’s core claims are sufficiently supported. In particular, the argument about the lack of semantic or logical role of intermediate tokens is still largely speculative and not backed by conclusive evidence. The discussion of research directions is interesting but does not substantially strengthen the central claim. Overall, our concerns about the strength and grounding of the main position remain, and thus our assessment is unchanged.

---

### Official Review · Reviewer_SWcw · 2026-03-11

**Significance:** 3
**Argument Clarity:** 3
**Rating:** 5
**Confidence:** 3

**Questions:**

Would effective non-linguistic intermediate tokens/representations validate your position if effective?

Do you believe that traces are useless for AI safety monitoring given your argument?

**Alternative Views Section:**

Yes

**Compliance With Llm Reviewing Policy A Conservative:**

Affirmed.

**Discussion Potential:**

4

**Final Justification:**

Updated due to rebuttal

**Paper Summary:**

The authors suggest that intermediate tokens are not "thinking" or "reasoning traces" and this view is actively harmful. This view is supported by some very recent works which show improved performance even though trained on incorrect traces (so the semantic content may not drive performance).

**Position:**

Yes

**Position In Title:**

Yes

**Related Work:**

3

**Strengths And Weaknesses:**

Timely and important concern. The anthropomorphising happening in research and by the public is indeed growing. The referred papers are the strongest support of the position.

I think the position that traces _never_ carry meaningful semantic information and such occurrences are merely accidental is going to be hard to defend. The position also doesn't seem to engage enough with recent work that does appear to help users and researchers through a reasoning-trace perspective.

**Support:**

3

---

> ### Author Rebuttal · Authors · 2026-03-30
>
> Thank you for your thoughtful review. Our responses follow
>
>
> **The position that traces *never* carry meaningful semantic information -** We note that our position is not that intermediate tokens can't ever have interpretable meaning but that they don't necessarily have to have any interpretable meaning. Any human interpretable meaning in the intermediate tokens might be a fortuitous coincidence of such rationales being present in the training data, rather than the model actually doing internal computations corresponding to those statements. In other words, the correlation here may be spurious rather than causal. This seems to be the clearest lesson to be drawn from the cited recent literature.
>
> Our position is bolstered by the facts that (1) there certainly isn't any theory drawing causal connections between the semantics of the intermediate tokens and the final solution--beyond the usual "intermediate tokens change the conditional distribution of the solution tokens" and (2) even the proponents of user interpretable semantics for intermediate tokens (c.f. Korbak et. al.) seem to realize this and write papers talking about why it is incumbent to "protect" the "fragile" connection between intermediate tokens and final solutions so as to allow for some kind of "monitorability" of LLMs (even if it may well be only illusory).
>
> Until there is evidence–beyond circumstantial–that LRM reasoning tokens correspond to internal computations, it seems unwise to depend on the intermediate tokens for “safety monitoring” in safety-critical domains. A third-party verification of the solutions/decisions seem to be the better way to go in such scenarios.
>
> At the least, we believe that our position should be aired as a countervailing view until someone actually shows the conditions under which we can ensure causal connection between intermediate token semantics and the LRM solutions--beyond mere change of conditional distribution--which can also be achieved without using the tokens that have meaning for end users (such as latent tokens).
>
> >Would effective non-linguistic intermediate tokens/representations validate your position if effective?
>
> Yes, effectiveness of non-linguistic intermediate tokens does indeed validate our position. In fact, as we discuss in the paper [page 8, lines 405-418, left colum], once we stop ascribing questionable interpretability to intermediate tokens, we can focus on training with intermediate tokens comprising vectors from the continuous embedding space. We also cite (Hao et al 2024) and (Zhang et al 2025a) that  show preliminary evidence on how such methods could improve solution accuracy.
>
> >Do you believe that traces are useless for AI safety monitoring given your argument?
>
> Until there is evidence–beyond circumstantial–that LRM reasoning tokens correspond to internal computations, it seems unwise to depend on the intermediate tokens for “safety monitoring” in safety-critical domains. A third party verification of the solutions/decisions seem to be the better way to go in such scenarios.
>
> As we noted, even the proponents of user interpretable semantics for intermediate tokens (c.f. Korbak et. al.)  seem to realize that the connection between intermediate tokens and internal computations may well be a case of spurious correlation acquired from training data,  and thus write papers talking about why it is incumbent to "protect" the "fragile" connection between intermediate tokens and final solutions so as to allow for some kind of "monitorability" of LLMs (even if it may well be only illusory).

---

> > ### Author Rebuttal · Reviewer_SWcw · 2026-04-03
> >
> > Thanks for the clarification. Indeed, I think it would be valuable to discuss the setting where reasoning tokens may not carry semantic information. I've raised my score, but remain uncertain.

---

### Official Review · Reviewer_SZLJ · 2026-03-15

**Significance:** 3
**Argument Clarity:** 3
**Rating:** 5
**Confidence:** 4

**Questions:**

1. Are the authors rejecting all forms of end-user interpretability, or only strong faithfulness-style interpretations? Do weaker but useful roles for trace inspection (e.g., debugging, safety monitoring) remain valid under this framework?
2. Since terms like "reasoning trace" are often used as convenient shorthand rather than literal cognitive claims, what specific terminology do the authors recommend the community adopt, and how would this shift concretely improve research methodology?

**Alternative Views Section:**

Yes

**Compliance With Llm Reviewing Policy A Conservative:**

Affirmed.

**Discussion Potential:**

4

**Final Justification:**

The rebuttal has fully addressed the initial concerns, and the paper remains a highly significant contribution to the current discourse on reasoning models. I am maintaining my score of 5 (Accept).

**Paper Summary:**

This position paper argues that the machine learning community should stop anthropomorphizing intermediate tokens generated by LLMs as genuine "reasoning" or "thinking" traces. By reviewing the intermediate-token generation research and surveying recent empirical evidence, the authors demonstrate that the semantic validity and human interpretability of these traces are often weakly correlated with final-answer correctness, and can even degrade under RL post-training. The paper advocates for a shift in perspective, suggesting researchers to treat these outputs as neutral derivational traces rather than trustworthy windows into model cognition, and to ground trust in verifiable solutions.

**Position:**

Yes

**Position In Title:**

Yes

**Related Work:**

4

**Strengths And Weaknesses:**

Strengths:

1. Highly relevant and impactful topic. The paper addresses a central concept in current LLM discourse, providing a clear normative position supported by a broad synthesis of recent evidence (e.g., weak correlation to correctness, robustness to noise, RL degradation).

2. Concrete downstream connections. The authors successfully connect abstract terminology issues to practical consequences, effectively detailing how anthropomorphic framing can mislead end users, create unwarranted trust, and motivate questionable research directions.

Weaknesses:

1. Indirect Empirical Support: The argument heavily relies on a literature synthesis of recent (and potentially unsettled) arXiv papers rather than new experiments on frontier models. The paper does not provide concrete evidence regarding the limits of extrapolating findings from controlled, algorithmic traces to broad natural-language reasoning models.

2. Overstated Contrast and Collapsed Concepts: The manuscript poses a strong rejection of anthropomorphic interpretations but often blurs the line between literal "thoughts" and practical interpretability. It fails to clearly distinguish unreliable global faithfulness from potentially valuable local diagnostic signals (e.g., debugging, monitorability, effort estimation).

**Support:**

3

---

> ### Author Rebuttal · Authors · 2026-03-30
>
> Thank you very much for your close read of our position and your overall positive assessment.
> Let us respond to your concerns about weaknesses, and your direct questions.
>
> **Indirect Empirical Support:** We would like to reassure the reviewer that our position was not just observational, but is also supported by our own direct experiments, some of which were covered in the literature synthesis. While the author anonymity restrictions prohibit us from explicitly pointing out which efforts were from us, once the double-blind restriction is lifted for the final version, we will clearly point out which works were done by our group vs. other groups, and we believe that will make the readers realize that our position is certainly motivated in part by our own direct studies.
>
> About experiments on frontier models, we note that some of the literature we review, such as Korbak et. al., and Bhambri et. al. do deal with frontier models and pretrained open source models. We also reiterate our point (page 7, line 378 left column, Harmony format) that with current frontier models, end users are not even getting the intermediate tokens/CoTs, but rather a summary/explanation generated in parallel by unknown means.
>
>
> >Q1. Are the authors rejecting all forms of end-user interpretability, or only strong faithfulness-style interpretations? Do weaker but useful roles for trace inspection (e.g., debugging, safety monitoring) remain valid under this framework?
>
> We note that our position  is not that intermediate tokens can't ever have interpretable meaning  but that they don't necessarily have to have  any interpretable meaning. Any human interpretable meaning in the intermediate tokens might be a  fortuitous coincidence of such rationales being present in the training data, rather than the model actually doing internal computations corresponding to those statements. In other words, the correlation here may be  spurious rather than causal. This seems to be the clearest lesson to be drawn from the cited recent literature.
>
> Our position is bolstered by the facts that (1) there certainly isn't any theory drawing causal connections between the semantics of the intermediate tokens and the final solution--beyond the usual "intermediate tokens change the conditional distribution of the solution tokens" and (2) even the proponents of user interpretable semantics for intermediate tokens (c.f. Korbak et. al.) seem to realize this and write papers talking about why it is incumbent to "protect" the "fragile" connection between intermediate tokens and final solutions so as to allow for some kind of "monitorability" of LLMs (even if it may well be only illusory).
>
> Until there is evidence–beyond circumstantial–that LRM reasoning tokens correspond to internal computations, it seems unwise to depend on the intermediate tokens for “safety monitoring” in safety-critical domains. A third-party verification of the solutions/decisions seem to be the better way to go in such scenarios.
>
> At the least, we believe that our position should be aired as a countervailing view until someone actually shows the conditions under which we can ensure causal connection between intermediate token semantics and the LRM solutions--beyond mere change of conditional distribution--which can also be achieved without using the tokens that have meaning for end users.
>
> >Q2. Since terms like "reasoning trace" are often used as convenient shorthand rather than literal cognitive claims, what specific terminology do the authors recommend the community adopt, and how would this shift concretely improve research methodology?
>
> While we prefer the neutral term **“intermediate tokens”** to refer to the tokens LLMs output before the solution, we should point out that our position is not against the “terminology” (however anthropomorphic), but against the subsequent anthropomorphization.  Many anthropomorphic terms in LLM literature, such as “thoughts”, “CoTs” have become too entrenched at this point to try to replace them. Our position is to caution the community that the anthropomorphization of actually  assuming that the  intermediate tokens must be telling the end user about LLM intermediate reasoning steps just because we **called them reasoning traces**.

---

> > ### Author Rebuttal · Reviewer_SZLJ · 2026-04-04
> >
> > I thank the authors for their detailed and thoughtful response. The clarification regarding the authors' own direct experiments and the inclusion of frontier model studies—particularly those addressing the lack of intermediate token transparency in current production models—sufficiently addresses the concerns regarding empirical grounding.I find the distinction between "fortuitous coincidence" and "causal computation" to be a compelling defense of the position.

---

### Decision · Program_Chairs · 2026-04-30

**Decision:**

Accept (regular)

**Comment:**

This position paper argues that the machine learning community should stop anthropomorphizing intermediate tokens generated by LLMs as genuine "reasoning" or "thinking" traces. Overall, the reviews agree that this is an important position and take-away message. Overall, however, the reviews are rather mixed, 2x accpets, 1x borderline reject and 1x reject. The two negative ones, however, may have slightly misunderstood the position. As clearly expressed in the authors' response, the main argument is not about whether the LLM intermediate tokens exhibit "human-like reasoning" which is clearly hard to pin down, but whether the reasoning can be said to lead to the solution in any logically interpretable sense. This is an important position and in the discussion with the reviewers, it beomce clear that the paper discussed also current evidence about how "wrong process" can still lead to "correct answer". So overall, I think the major downsides have been clarified.